# Herald Patch as the Only Evidence of Pityriasis Rosea: Clinical, Laboratory and Pathogenetic Features

**DOI:** 10.3390/v17010119

**Published:** 2025-01-16

**Authors:** Francesco Drago, Astrid Herzum, Serena Varesano, Gaetano Serviddio, Francesco Broccolo, Giulia Ciccarese

**Affiliations:** 1Dermatology Clinic, Villa Montallegro Health Clinic, Via Monte Zovetto 27, 16145 Genoa, Italy; 2Section of Dermatology, IRCCS Istituto Giannina Gaslini, Via G. Gaslini 5, 16147 Genoa, Italy; astridherzum@yahoo.it; 3Hygiene Unit, IRCCS Ospedale Policlinico San Martino, Largo R. Benzi 10, 16132 Genoa, Italy; 4Internal Medicine, Liver Unit, C.U.R.E. (University Centre for Liver Disease Research and Treatment), Department of Medical and Surgical Sciences, University of Foggia, Viale Pinto 1, 71122 Foggia, Italy; gaetano.serviddio@unifg.it; 5Department of Experimental Medicine, University of Salento, Piazza Tancredi 7, 73100 Lecce, Italy; francesco.broccolo@unisalento.it; 6Section of Dermatology, Department of Medical and Surgical Sciences, University of Foggia, Viale Pinto 1, 71122 Foggia, Italy; giulia.ciccarese@unifg.it

**Keywords:** pityriasis rosea, herald patch, human herpesvirus-6 (HHV-6), human herpesvirus-7 (HHV-7), atypical exanthems, viral reactivation, immunological response, cutaneous lesions, cytokine profile, differential diagnosis

## Abstract

Pityriasis rosea (PR) is a self-limited exanthem associated with the endogenous systemic reactivation of human herpesvirus (HHV)-6 and HHV-7. The disease typically begins with a single erythematous patch on the trunk (herald patch), followed by a secondary eruption of smaller papulosquamous lesions. Rarely, the herald patch may be the only cutaneous manifestation of PR. The present work aimed to examine the clinical and laboratory features of the PR cases characterized by the herald patch as the sole cutaneous manifestation and to compare them with the classic form of PR. An observational, retrospective study was conducted on patients presenting with herald patch as the only sign of PR (cases) and on a series of age- and sex-matched patients who presented with a typical PR pattern (controls). The records of the patients were extracted from a PR registry, which collected data on patients with PR diagnosed from 2003 to 2023 by at least two dermatologists from the same study team. Nineteen patients (eleven males, eight females) with a mean age of 27.1 years presented the herald patch as the only cutaneous manifestation of PR. Nineteen age- and sex-matched patients were considered controls. In the cases, the exanthem duration was shorter than in controls, and the mean HHV-6 and HHV-7 DNA plasma load was lower than in controls. This rare variant of PR might be considered an abortive form of the exanthem that occurs when the HHV-6/7 reactivation from latency is contrasted by a more robust immunological response than in classic PR.

## 1. Introduction

Pityriasis rosea (PR) is an acute, self-limited exanthematous disease that primarily affects children and young adults between 10 and 35 years of age. The disease is widespread across disparate populations, with a global incidence of 0.64 per 100 dermatological patients and a slight female prevalence [1,2,3]. A large body of evidence supports the causal role of the endogenous systemic active human herpesvirus (HHV)-6 and HHV-7 infections in PR pathogenesis. The cytopathic effects specific to HHVs, such as ballooning cells and syncytia formation, were observed in the culture of peripheral blood mononuclear cells (PBMCs) from PR patients, the detection of cell-free HHV-6 and HHV-7 DNA in the plasma of PR patients, and the presence of their viral antigens and messenger RNA expression in PR skin lesions are all markers of active viral replication [1,4,5]. Furthermore, HHV virions in various stages of morphogenesis have been detected by electron microscopy in skin lesions and the supernatant of co-cultured PBMCs from PR patients [6]. Lastly, the levels of several interleukins (ILs) and chemokines, such as IL-17, IL-22, IL-36, IFN-*γ*, vascular endothelial growth factor (VEGF), CX3CL1/fractalkine, and CXCL10, were found to be upregulated in the sera of PR patients compared to controls. These cytokines and chemokines networks are further hints that PR is associated with activation of cellular immunity and induction of an inflammatory response against a virus [7,8,9].

After the primary infection, which is usually asymptomatic or may cause the exanthema subitum (roseola) or a febrile illness without any rash and rarely accompanied by convulsions, HHV-6 and HHV-7 remain lifelong latent in the body, mainly associated with PBMCs and in salivary glands [1]. Several mechanisms may trigger HHV-6 and HHV-7 reactivation and, therefore, the occurrence of PR: exhaustion, post-surgery or post-traumatic stress, fever, states of immunosuppression, pregnancy, and inappropriate exposure to ultraviolet rays [1].

In fact, during the COVID-19 pandemic, PR and herpes zoster have been frequently reported. The psychological stress linked to the pandemic and the immunosuppression associated with SARS-CoV-2 infection may enable the reactivation of latent viral infections such as HHV-6 and HHV-7, which cause PR manifestations, or the Varicella–Zoster virus, which causes herpes zoster.

SARS-CoV-2 infection decreases the number and efficiency of different T lymphocyte populations, particularly CD4+ T cells, CD8+ T cells, and natural killer cells. CD4+ T lymphocytes play a central role in initiating, coordinating, and maintaining the antiviral immune response, and CD8+ T cells are also significant players in virus control [1,10]. Conversely, Th17 cells, IL17A, and the Th17/T regulatory (Treg) cell ratio increase in the peripheral blood of patients with COVID-19 [10,11,12]. It has been demonstrated that an imbalance between Th17/Treg cells and an increased level of IL17 may facilitate reactivation and lytic infection of HHVs [10,11,12,13,14]. In addition to creating a dysregulated Th17 immune response, SARS-CoV-2 infection may have a direct trans-activating role, triggering HHV-6 and HHV-7 reactivation and causing, indirectly, the onset of PR [10,15].

Recently, it has been suggested that Toll-like receptors (TLRs) may play a role in the pathogenesis of PR. El-Ela et al. found a significantly higher expression of TLR 2, 3, 4, 7, 8, and 9 in the lesional skin of PR patients and a significantly higher viral load for both HHV-6 and HHV-7 in PR patients compared to the controls [16]. TLR expression was also increased in PBMCs of PR patients compared to controls [17]. It must be emphasized that TLRs 3, 7, 8, and 9 are involved in the responses against several viral infections [18]. Therefore, it is plausible that following the HHV-6 and HHV-7 systemic reactivation and their arrival in the skin, an increased expression of TLRs can be induced (both systemically and in the skin lesions of patients with PR). TLRs, in turn, can mount an immune response directly through the induction of cytokines and other pro-inflammatory mediators or indirectly increase the number of Langerhans cells in the PR lesions. Langerhans cells can participate in the innate and adaptive immune response by releasing inflammatory chemokines and stimulating T cells.

Increased TLRs are seen in virtually any inflammatory condition (lupus, psoriasis, acne, other infections, drug reactions) and are not specific. However, the increased TLRs in PBMCs [16] may suggest that PR is a systemic viral disease with primary and early burden on the immune system.

Differently from PR, PR-like eruption (PR-LE) is a drug- or vaccine-induced skin rash with clinical features that strikingly resemble genuine PR. Although it is often hard to clinically distinguish them, criteria have recently been proposed for distinguishing between the two forms [10,19]. PR-LE has an entirely different pathogenesis because, unlike PR, it is a delayed hypersensitivity response to a vaccine or a drug. Roughly, one can say that PR-LE can be compared to PR just as morbilliform drug eruptions are to measles. PR-LE has been frequently reported during the COVID-19 pandemic and following vaccination against COVID-19, but in contrast to genuine PR, PR-LE is usually not associated with HHV-6 and/or HHV-7 systemic reactivation. A possible pathogenetic mechanism for PR-LE is molecular mimicry between a viral epitope (i.e., SARS-CoV-2 spike glycoprotein), a viral vector, or a vaccine ingredient and host proteins that could result in a T-cell-mediated hypersensitivity skin reaction [10].

Clinically, PR is characterized by prodromal symptoms such as mild fever, sore throat, headache, loss of appetite, and gastrointestinal and upper respiratory symptoms that may precede or be associated with the exanthem in up to 69% of the patients [1,3].

Pruritus is usually mild or absent [1]. The onset of PR is characterized by a single, erythematous macule or papule mainly on the trunk that enlarges over a few days, developing a medallion-like, scaly patch or plaque (“mother patch” or “herald patch”). This initial lesion has a depressed center and a raised border with a “collarette” of scales attached to the periphery with its free edge extending internally [1,2,3]. The herald patch (HP), which is usually asymptomatic, is followed after about two weeks by a secondary eruption consisting of smaller papulosquamous lesions along the cleavage lines of the skin in a distribution that resembles a “Christmas tree” or “theatre curtain” pattern on the trunk, mainly when it affects the back [1,3]. The face, scalp, palms, and soles are usually spared, and maculopapular, erythematovesicular, and petechial oropharyngeal lesions may be present (28% of cases) [20]. Differential diagnoses include secondary syphilis, nummular eczema, guttate psoriasis, tinea corporis, pityriasis versicolor, pityriasis lichenoides chronica, and atypical exanthems due to Epstein Barr virus, Parvovirus B19, [1,3,21,22,23,24] or to other virus infection/reactivation [1,3].

According to the recently described criteria, PR-LE should be carefully distinguished from PR [19]. In typical PR cases, all the cutaneous lesions gradually heal in 2–6 weeks without leaving any sequelae. Therefore, the best treatment is to reassure the patient by recommending rest. Atypical PR cases associated with extensive, persistent lesions and systemic symptoms can impact the patient’s quality of life, and, therefore, treatment may be prescribed. A recent meta-analysis showed that acyclovir is the best option for these cases [25].

The prevalence of these atypical variants of PR may be underestimated. PR may be defined as atypical for the lesions’ morphology, size, number, location, and distribution and for the severity of associated symptoms and course [15,16,17,18,19]. Among the rare, atypical forms of PR, there are cases in which HP may be the only manifestation of the eruption [1,26,27,28,29]. The clinical and laboratory features of PR characterized by the HP without the secondary eruption have been poorly investigated. The present work aimed to investigate the clinical and laboratory features of PR with HP as the unique manifestation of the disease and to compare this rare variant of PR with the classic one.

## 2. Patients and Methods

An observational, retrospective case-control study was conducted on patients presenting with HP as the only sign of PR and on a series of age- and sex-matched patients who presented with a typical PR pattern. The records of the patients described in the present study were extracted from a PR registry, as previously described [30], which collected data of PR patients diagnosed clinically and independently from 2003 to 2023 by at least two dermatologists from the same study team. Notably, one of them (F.D.) diagnosed all patients. Histopathology supported the diagnosis whenever needed. To avoid misdiagnoses, a mycological analysis for direct microscopic examination with KOH and mycological culture and serological tests such as *Treponema pallidum* haemagglutination assay (TPHA) and antibodies against human immunodeficiency virus (HIV) were performed in all cases of PR with atypical manifestations. Blood samples were taken from all selected patients at the first visit to determine the titer of the antibodies against HHV-6 and HHV-7 and to search for HHV-6 and HHV-7 DNA loads in plasma by calibrated quantitative real-time PCR (CQ-PCR), as previously described [5]. More specifically, CQ-PCR assays were performed on an ABI PRISM 7700 SDS (Applied Biosystems, Foster City, CA, USA). Four independent TaqMan real-time PCR assays were executed to detect HHV-6, HHV-7, calibrator, and a fragment of the single-copy human CCR5 gene; this system was used to quantify the content of human genomic DNA (copies/mL) both in plasma and in the PBMCs. At least 1 mg of genomic DNA recovered from each PBMC pellet was subjected to PCR analysis [5].

The first visit and blood test occurred within ten days of PR onset in all the patients. Data were reviewed to examine the gender and age of patients, the site of the herald patch, the duration of the exanthem, the presence of mucosal lesions, associated systemic symptoms, previous infections, drug intake before eruption, and laboratory investigations. Only subjects who were otherwise healthy, immunocompetent, and did not take any medication before and during the eruption were included in the present study. Drug-induced PR-like eruptions were carefully excluded based on clinical and pharmacological history and according to recently proposed criteria [19]. All the patients gave informed consent to take blood samples for laboratory investigations, collect their data, and be included in this study. This observational retrospective study adhered to the ethical standards of the institutional and national research committees, the 1964 Helsinki Declaration, and its later amendments. The medical ethical committee of the University of Genoa approved the study (Registry number 163/2020).

### Statistical Analysis

Data were statistically described in terms of mean ± standard deviation (SD), frequencies (number of cases), and relative frequencies (%) when appropriate. Comparisons between groups were done using the *t*-test for normally distributed quantitative variables and the chi-square (χ^2^) test for categorical data. Correlation between various variables was done using the Pearson moment correlation equation for linear relation. *p* < 0.05 was considered statistically significant.

## 3. Results

Between January 2003 and December 2023, we collected data from 964 patients with different forms of PR. Twenty-seven patients presented HP as the only evidence of PR (2.8%). Still, we recruited only nineteen of them (2%), eleven males (57.8%) and eight females (42.2%) with a mean age of 27.1 years, who had completed all microbiological investigations (cases, Table 1).

To exclude other differential diagnoses, all these patients performed mycological analysis and serology for T. pallidum and HIV infection [1], obtaining negative results. A histopathological study was conducted on three out of nineteen patients, and it was compatible with the PR diagnosis [1]. Considering age- and sex-matched patients (controls, Table 2), 38 patients with classic PR, as represented in Figure 1, were finally included in the study.

In most patients (32/38, 84%), the herald patch involved the trunk (18/19, 95% of the cases; 14/19, 74% of the controls); only 6/38 (16%) of the enrolled patients presented the herald patch on the thigh: 1/19 (5%) of the cases (Figure 2) and 5/19 (26%) of the controls.

Concerning the site of the HP, no statistical significance was found at the χ^2^ test with Yates correction for the two groups (*p* = 0.181996). Nearly all patients, 33/38 (87%), had systemic symptoms associated with the skin eruption: 16/19 (84%) of the cases and 17/19 (89%) of the controls. Only 5/38 (13%) of the enrolled patients did not complain of systemic symptoms: 3/19 (16%) of the cases and 2/19 (11%) of the controls. No statistical significance was found at the χ^2^ test with Yates correction (*p* = 1).

Mucosal oropharyngeal lesions were found to be associated with skin lesions in 11/38 (29%) of the overall patients: 3/19 (16%) of the cases and 8/19 (42%) of the controls. Therefore, the majority of patients (27/38, 71%) had no enanthem: 16/19 (84%) of the cases and 11/19 (58%) of the controls. No statistical significance was found at the χ^2^ test with Yates (*p* = 0.152493).

The cutaneous eruption lasted, on average, 32 days, with a mean duration of 26.5 days among the cases and 37.5 days among the controls. The cases showed a shorter duration of the exanthem than the controls; this difference was highly statistically significant at the *t*-test (*p* < 0.00001).

Concerning the laboratory investigations, the mean HHV-6 viremia detected in plasma was 25 copies/mL, with mean values of 18 copies/mL among the cases and 32 copies/mL among the controls. The cases showed a significantly lower HHV-6 viral load in plasma than in the controls; the difference between the groups was statistically significant at the *t*-test (*p* < 0.00001). The mean HHV-7 viral load detected in plasma was 11 copies/mL, with mean values of 8 copies/mL among the cases and 13 copies/mL among the controls. The cases revealed a significantly lower HHV-7 DNA plasma load than the controls; the difference between the groups was statistically significant at the *t*-test (*p* = 0.007275). Regarding serology, the mean HHV-6 IgG antibody titer was 1:160 in the cases group and 1:80 in the controls group. The mean HHV-6 IgM antibody titer was 1:80 in the cases group and 1:160 in the controls group. At the *t*-test, no statistical significance between the two groups was found (*p* = 0.20687 and *p* = 0.180151, respectively). The mean HHV-7 IgG antibody titer was 1:80 in the cases group and 1:80 in the control group. The mean HHV-7 IgM antibody titer was 1:80 in the cases group and 1:80 in the control group. At the *t*-test, no statistical significance between the two groups was found (*p* = 0.420163 and *p* = 0.172066, respectively).

## 4. Discussion

Several review articles about the clinical variants of PR have reported the possibility that the disease may present only with the HP not followed by the secondary eruption [1,27,28,31]. However, this rare form of PR has not yet been described in case reports, case series, or original articles. The present study is the first to investigate the clinical and laboratory features of this atypical form of PR and compare it with the classic PR. Clinically, when PR manifests only with HP, the differential diagnoses that should be considered are quite different from the classic PR. The HP is a single lesion, whereas PR is a diffuse eruption. The differential diagnoses for PR with only HP include tinea corporis, nummular eczema, erythema annulare centrifugum [31], mycosis fungoides [32], and localized secondary syphilis [33]. A helpful distinguishing feature among these conditions is the location of the scale within the erythematous lesion. The PR with only HP and erythema annulare centrifugum are characterized by non-indurated, annular patches with trailing scales (collarette of scales) attached to the lesion’s periphery and extending internally. However, erythema annulare centrifugum tends to be recurrent and is not associated with constitutional symptoms [34]; conversely, PR may rarely relapse (3.7%) [30], and the presence of systemic symptoms is more frequent (69%) [1]. In tinea corporis, the edge of the scale appears to lead the erythema in an annular or polycyclic pattern; the border advances centrifugally while the center of the lesion may exhibit clearing. Furthermore, follicular pustules and even vesicles may occur; a mycological examination can be decisive in case of doubt, especially when tinea incognita is suspected [35,36]. Nummular eczema, differently from the HP of PR, is characterized by scales that cover the entire erythematous patch/plaque without central clearing [21]. Mycosis fungoides has a clinical presentation resembling several different inflammatory dermatoses, especially in the early stages, and often a biopsy with histological examination is necessary to make a diagnosis. The overlapping clinical features of mycosis fungoides with other benign dermatoses and the frequent application of immunosuppressing medications that may diminish the intensity of the lesion often cause a delay in diagnosis [32]. However, the HP of PR can be easily recognized because it has a limited duration in time and spontaneous healing, whereas mycosis fungoides has a prolonged clinical course. Annular-arciform symmetric localized lesions have also been described as the only manifestation of secondary syphilis, especially in cases of *T. pallidum* reinfection [33]. Annular-arciform lesions in syphilis can involve every site, mainly the anogenital area. Although the HP of PR has already been described in the genitalia, this location is unusual since the trunk is preferred, followed by the neck and proximal extremities [27]. Performing a serological test for syphilis and, if available, PCR tests for detecting *T. pallidum* DNA in skin lesions [37,38] can clarify the doubts.

Our study did not show significant clinical and serological differences between patients with classical PR and HP alone. Indeed, regarding the site of the HP, the presence of mucosal lesions, associated systemic symptoms, and the titer of the antibodies against HHV-6 and HHV-7, there were no statistically significant differences. On the contrary, the findings concerning the duration of HP and HHV-6 and HHV-7 viral load in the plasma compared to the eruption of classical PR have brought interesting results to advance hypotheses on the pathogenesis of this atypical form of PR.

Generally, the pathogenetic mechanisms that underlie the different forms of PR (namely classic, relapsing, persistent, and pediatric PR) depend on the different interactions between the systemic activation of HHV-6 and/or HHV-7 and the host’s immunological response [1,3,10]. In the classic PR, the HHV-6 and/or HHV-7 systemic reactivation is promptly controlled by the patient’s immune system, and the disease follows its normal course. Relapsing PR is a variant of the exanthem that recurs, usually within 1 year from the initial presentation, due to a temporary drop in the patient’s cell-mediated immune surveillance [30]. However, the immune system is not completely ineffective, though it is still recovering from the failure that permitted the primary viral reactivation. This explains why PR relapsing eruption and systemic symptoms are usually less severe than during the primary episode. Likewise, patients may suffer “relapses” within a few months, even in primary infections caused by other viruses [39]. A persistent HHV-6 and/or HHV-7 reactivation with higher viral loads than in the typical PR accounts for the unusual persistence of the disease characterized by more lasting skin and oral lesions and more frequent and severe systemic symptoms compared to the classic PR [40]. Pediatric PR may be considered a distinct form due to its peculiar clinical and laboratory features in which the oropharyngeal lesions are more common than in adults, the mean time lapse between HP and the generalized eruption is shorter (4 days versus about 2 weeks), and the exanthem duration is shorter as well [40]. The HHV-6/7 average plasma viremia, both during the acute phase and after recovery, is higher in children than adults since the primary HHV-6/7 infection, which occurs most commonly under 3 years, has just been overcome. Therefore, the immune system must promptly regain control of the active viral infection and systemic symptoms, and PR duration is shorter than in adults.

The pathogenic mechanism underlying the manifestation of PR with HP as the only sign of the disease has never been investigated before. The findings of our study may suggest some pathogenic mechanisms that could explain the clinical differences between this form of atypical PR and classical PR. We found that when PR manifests with only the HP, it has a shorter exanthem duration than the classic PR, and the mean HHV-6 and HHV-7 DNA load in plasma was lower than the controls with classic PR. Remarkably, all these differences reached a high statistical significance.

Therefore, an attempt to explain the pathogenesis of PR characterized only by the HP is the hypothesis that it could be considered an abortive form of the disease that happens when the HHV-6 and/or HHV-7 reactivation from latency is countered by an immunological response (mainly orchestrated by T CD4+ and T CD8+ lymphocytes [1,3,20]) that is more effective than in classic PR.

In conclusion, PR that occurs only with HP is a rare disease variant that may resemble other inflammatory, infectious, or neoplastic dermatoses, quickly leading to misdiagnoses. The diagnosis remains mainly clinical, as in the cases of classic PR. Laboratory investigations and histopathology are advisable in doubtful cases.

Dermatologists should be aware of this rare variant of PR and be well-trained to consider other possible differential diagnoses.

## Figures and Tables

**Figure 1 viruses-17-00119-f001:**
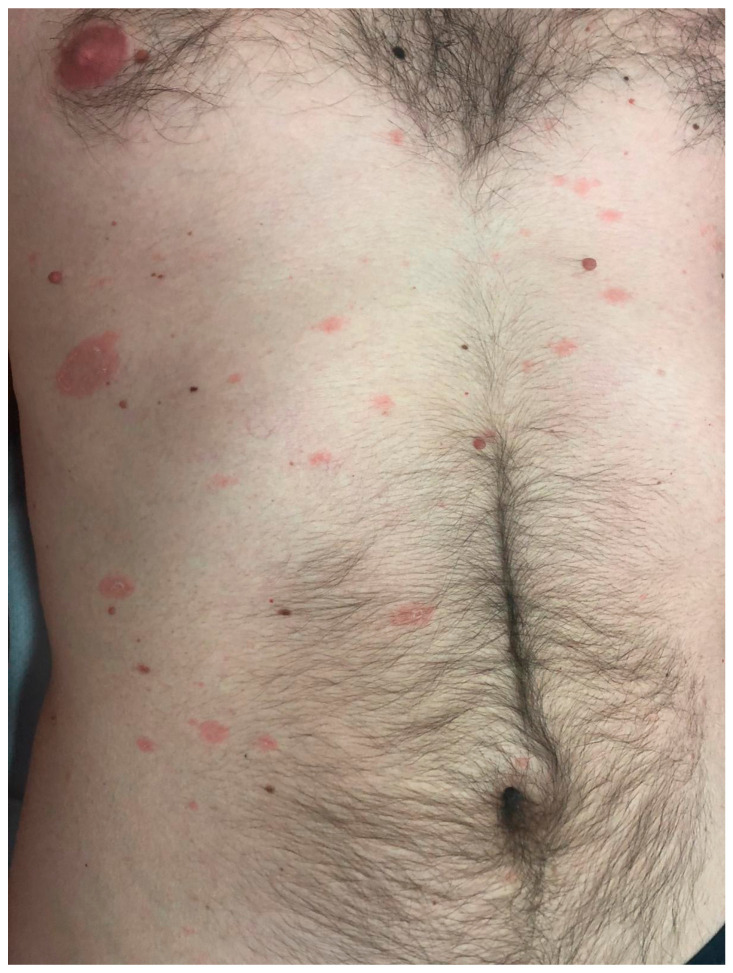
Classic pityriasis rosea: erythematous, round, scaly plaque in the right subpectoral region (herald patch); smaller papulosquamous lesions along the cleavage lines of the trunk (secondary eruption).

**Figure 2 viruses-17-00119-f002:**
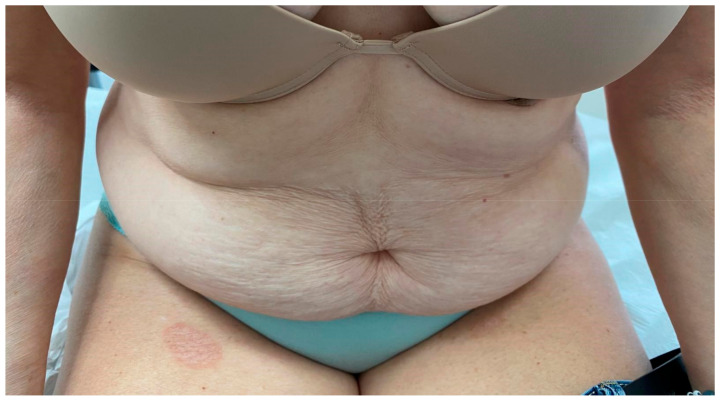
Pityriasis rosea characterized only by the herald patch: erythematous, oval, scaly plaque on the dorsal surface of the right tight; absence of other cutaneous lesions.

**Table 1 viruses-17-00119-t001:** Clinical and laboratory features of the patients with HP as the only evidence of PR ^1^.

Patient n°	Age	Sex	Herald Patch Site	Eruption Duration Days	Mucosal Lesions	Systemic Symptoms	Sampling Days (From the Onset of the Rash)	DNA Copies/mL Plasma HHV-6	DNA Copies/mL Plasma HHV-7	HHV-6 Antibody Titer IgG	HHV-6 Antibody Titer IgM	HHV-7 Antibody Titer IgG	HHV-7 Antibody Titer IgM
1	26	m	trunk	32	no	fatigue, headache, sore throat	10	15	<10	1/80	1/40	1/80	NEG
2	19	f	trunk	28	no	fatigue, irritability	6	23	12	1/320	1/80	1/80	NEG
3	31	m	thigh	25	no	fatigue, lack of appetite	12	10	<10	1/80	1/80	1/40	NEG
4	18	f	trunk	20	erythematous papules, petechiae	fatigue, sore throat, irritability, difficulty concentrating	8	20	10	1/160	1/160	1/80	1/40
5	23	f	trunk	30	no	fatigue, lack of appetite	7	18	NEG	1/160	1/80	1/40	NEG
6	29	f	trunk	28	petechiae, strawberry tongue	headache, irritability	10	15	15	1/320	1/80	1/80	1/80
7	34	m	trunk	20	no	no	12	10	NEG	1/160	1/80	1/40	NEG
8	30	m	trunk	25	no	fatigue, lack of appetite	8	12	NEG	1/320	1/80	1/80	1/40
9	26	m	trunk	28	no	sore throat	10	18	15	1/160	1/80	1/160	1/80
10	20	m	trunk	25	no	fatigue	8	18	10	1/80	1/320	1/160	1/80
11	17	f	trunk	30	erythematous papules, vesicles	fatigue, irritability, difficulty concentrating	8	25	12	1/80	1/320	1/80	1/40
12	28	m	trunk	27	no	headache, irritability	7	30	10	1/160	1/160	1/40	1/40
13	31	m	trunk	25	no	fatigue	10	15	<10	1/80	1/160	1/40	NEG
14	29	f	trunk	25	no	no	10	18	<10	1/80	1/80	1/80	1/80
15	34	m	trunk	28	no	sore throat	8	12	15	1/160	1/40	1/80	1/40
16	30	m	trunk	30	no	no	10	25	10	1/80	NEG	1/80	NEG
17	24	f	trunk	24	no	fatigue, insomnia	10	15	<10	1/160	1/160	1/160	1/40
18	35	m	trunk	30	no	fatigue, lack of appetite	7	20	12	1/80	1/40	1/80	1/40
19	31	f	trunk	24	no	fatigue, irritability, difficulty concentrating	10	15	<10	1/80	1/80	1/80	NEG

^1^ HHV-6 and HHV-7 DNA copies/mL in plasma range from 0 (negative) to ≥30; antibody titers range from 0 (negative) to 1/320.

**Table 2 viruses-17-00119-t002:** Clinical and laboratory features of patients with classic PR (controls) ^2^.

Patient n°	Age	Sex	Herald Patch Site	Eruption Duration Days	Mucosal Lesions	Systemic Symptoms	Sampling Days (From the Onset of the Rash)	DNA Copies/mL Plasma HHV-6	DNA Copies/mL Plasma HHV-7	HHV-6 Antibody Titer IgG	HHV-6 Antibody Titer IgM	HHV-7 Antibody Titer IgG	HHV-7 Antibody Titer IgM
1	18	m	trunk	38	no	no	8	35	15	1/80	1/320	1/40	1/160
2	28	f	thigh	35	erythematous maculo-papules	fatigue, headache	7	30	10	1/160	1/160	1/80	1/80
3	32	m	trunk	42	macules, petechiae	fatigue, lack of appetite	10	26	<10	1/160	1/320	1/40	1/40
4	19	f	trunk	38	no	fatigue, pruritus	8	46	18	1/80	1/160	1/40	NEG
5	30	m	trunk	35	no	no	7	35	15	1/80	1/80	1/160	1/80
6	31	m	thigh	40	no	fatigue, irritability, headache	10	20	<10	1/40	1/160	1/40	NEG
7	25	f	trunk	36	no	fatigue, sore throat	10	15	15	1/160	1/320	1/80	1/80
8	32	m	trunk	38	erythematous macules and vescicles	fatigue, irritability, difficulty concentrating	6	28	18	1/320	1/160	1/160	1/160
9	30	f	thigh	48	erythematous papules and macules	fatigue, irritability, insomnia	10	36	10	1/80	1/320	1/40	1/40
10	28	m	trunk	40	no	headache, arthralgia, mialgia	8	32	15	1/160	1/80	1/80	NEG
11	22	f	trunk	35	erythematous macules and papules	fatigue, irritability	10	56	20	1/80	1/160	1/40	1/80
12	33	m	trunk	32	no	fatigue, sore throat	6	42	<10	1/80	1/320	1/40	1/40
13	21	f	trunk	40	erythematous macules and papules	fatigue, irritability, difficulty concentrating	8	35	15	1/40	1/80	1/40	NEG
14	21	f	trunk	38	no	fatigue, lack of appetite	10	46	<10	1/80	1/80	1/80	1/40
15	32	f	trunk	35	macules, papules, petechiae	fatigue, headache, irritability	8	26	12	1/40	1/80	1/80	1/40
16	32	m	trunk	30	no	fatigue	10	20	15	1/80	1/80	1/40	1/40
17	28	m	trunk	35	no	fatigue, pruritus, insomnia	8	18	10	1/80	1/160	1/40	NEG
18	27	m	thigh	35	erythematous macules, petechiae	fatigue, headache	10	36	12	1/80	1/160	1/80	1/40
19	29	m	thigh	42	no	fatigue, difficulty concentrating, irritability	8	20	18	1/80	1/80	1/40	1/80

^2^ HHV-6 and HHV-7 DNA copies/mL in plasma range from 0 (negative) to ≥30; antibody titers range from 0 (negative) to 1/320.

## Data Availability

Data are available from the corresponding authors upon reasonable request.

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
