# Peer review of "Herald Patch as the Only Evidence of Pityriasis Rosea: Clinical, Laboratory and Pathogenetic Features"

_viruses, 2025, doi:10.3390/v17010119_

Round 1
Reviewer 1 Report
Comments and Suggestions for Authors
|
This is a well-presented study that will be of interest to the dermatology and virology communities.
Lines 57-59: "These cytokines and chemokine networks are further evidence..." and Lines 94 - 96: "The increased TLRs in PBMCs may confirm that PR is a systemic viral disease..." While I agree with the authors that the accumulated evidence supports that PR is caused by HHV reactivation, an increase in cytokine or TLR expression in PBMCs is circumstantial evidence at best. Increased TLRs are seen in virtually any inflammatory condition (lupus, psoriasis, acne, other infections, drug reactions) and are not specific. Unless you are doing an ELISpot or similar assay against HHV antigens, I would not say that T cell activation, cytokine elaboration, or TLR expression "confirms" anything other than immune activation. The other data on HHV reactivation including plasma DNA and viral cytopathic change are more supportive of that specific etiology, but the phrasing in your introduction seems to suggest that cytokine and TLR expression are somehow strongly supportive of a viral cause... there is speculation in that connection that should be more clearly acknowledged. Line 200: "No statistical significance was found at χ2 test with Yates correction..." Can you clearly state the hypotheses being tested here? Is it location of the patch? Sorry if I missed it, but did not seem clear. Regarding your conclusion: HHV is also reactivated in many patients in the intensive care unit who have sepsis for other reasons... Any significant systemic illness can cause HHV reactivation, and SARS-CoV-2 is an illustration. But only a small fraction of patients with HHV reactivation get PR (and typically it is those who are young and healthy, and not those who are critically ill or immune suppressed). While a more active immune response may explain the abortive HP-only cases that you report, an intact immune response to HHV (as seen in young, healthy patients) may itself be part of (or required) for the pathogenesis of PR. So the relationship may not be as simple as less immune response leads to worse or more generalized PR. Please comment on this if you feel it is appropriate.
|
Reviewer 2 Report
Comments and Suggestions for Authors
The manuscript is overall well written, concise, and addresses an interesting question regarding the clinical and laboratory differences between classic pityriasis rosea (PR) and PR that presents only with a herald patch and not with the more diffuse exanthem that the former entails. Hearld patch-only PR is not well described in the literature, so the paper has merit for that reason alone. Furthermore, the inclusion criteria are quite rigorous and appropriately exclude any confounding variables, the methods are sound, and statistical tests are appropriately applied. The tables are formatted well, and the figures are high quality. I have only minor comments, which are listed below.
1. Please change instances of "unique" to "only cutaneous" or "sole cutaneous" in the abstract for clarity (page 1 lines 24 and 26).
2. Please include reference ranges for plasma HHV6/7 DNA as well as antibody titers somewhere in the manuscript (footnotes in tables would be most appropriate).
3. Although the content is easily understood by a native American English speaker, some minor editing could be performed to more stringently comply with standard English usage.
Thank you for the opportunity to review this interesting manuscript.
Reviewer 3 Report
Comments and Suggestions for Authors
ACCEPT GOOD PAPER NY THE BEST EXPERT
Reviewer 4 Report
Comments and Suggestions for Authors
The authors of Herald Patch as the Only Evidence of Pityriasis Rosea: Clinical, 2 Laboratory and Pathogenetic Features paper, are presenting a very comprehensive study in which they investigate the clinical and laboratory features of Pityriasis rosea with herald patch (HP), as the unique manifestation of the disease and to compare this rare variant of Pityriasis rosea with the classic one.
The authors collected data from 964 patients with different forms of PR.
Blood samples were taken in all selected patients at the first visit to determine the titre of the antibodies against HHV-6 and HHV-7 and to search for HHV-6 and HHV-7 DNA loads in plasma by calibrated quantitative real- time PCR (CQ-PCR), as previously described [5].
The cited paper is from 2005, and even if there are common authors, please give some information about the reagents, qPCR platform used for viral load assessing.
5 Broccolo F, Drago F, Careddu AM, Foglieni C, Turbino L, Cocuzza CE, Gelmetti C, Lusso P, Rebora AE, Malnati MS. Additional 340 evidence that pityriasis rosea is associated with reactivation of human herpesvirus-6 and -7. J Invest Dermatol. 2005 341 Jun;124(6):1234-40.
The laboratory data are sustained by 2 illustrative figures of Pityriasis rosea.
